# Analysis of the Spatial–Temporal Pattern of the Newly Increased Cultivated Land and Its Vulnerability in Northeast China

Guoming Du [1], Xiaoyang Wang [1,*], Jieyong Wang [2], Yaqun Liu [2] and Haonan Zhang [2,3]

1   School of Public Administration and Law, Northeast Agricultural University, Harbin 150030, China; duguoming@neau.edu.cn
2   Institute of Geographic Sciences and Natural Resources Research, Chinese Academy of Sciences, Beijing 100101, China; wjy@igsnrr.ac.cn (J.W.); liuyaqun@igsnrr.ac.cn (Y.L.); zhanghaonan2538@igsnrr.ac.cn (H.Z.)
3   University of Chinese Academy of Sciences, Beijing 100049, China
*   Correspondence: s201201012@neau.edu.cn

**Abstract:** Ensuring compliance with China's "1.8 billion mu" (120 million hectares) cultivated land preservation policy is a fundamental goal of land policy. Northeast China has experienced significant cultivated land expansion due to rigorous compensation policies over the past two decades, resulting in sustainable increases in grain output. This research employs remote sensing data to examine the spatial–temporal pattern and vulnerability of newly increased cultivated land expansion in Northeast China and its potential impact on food security. Results indicate a 3.08% increase in newly increased cultivated land from 2000 to 2020, with the majority located in the Sanjiang Plain's humid area and Inner Mongolia's arid and semi-arid regions. The low quality of newly added cultivated land makes it highly vulnerable. Temperature instability significantly and negatively correlates with cultivated land expansion. The vulnerability of cultivated land is negatively and significantly related to grain yield, suggesting an adverse impact on national food security. This study focuses on the marginal impact of newly increased cultivated land and proposes policy recommendations.

**Keywords:** newly increased cultivated land; spatial–temporal pattern; cultivated land vulnerability; food security

## 1. Introduction

In response to the growing global population and the desire for an improved quality of life, cultivated land has been expanded and intensified worldwide [1]. China, with only 10% of the world's cultivated land, feeds approximately 22% of the global population. To enhance its food production capacity, the Chinese government has implemented several policies aimed at preserving cultivated land [2,3]. Although these policies have helped to stabilize the overall quantity of cultivated land in China, significant regional differences persist [4]. China's northeast region is a significant contributor to the country's primary grain production area, with its cultivated land experiencing growth in recent years.

Northeast China is a significant region for grain production, with its output accounting for a quarter of China's total grain output, which can support approximately 100 million people. However, the pressure on grain support in this region has increased due to the continual reduction of cultivated land along the southeast coast in recent years [5,6]. To guarantee national food security, the Chinese government has introduced measures to enhance grain production in Northeast China. In 2021, the Chinese government made black soil preservation a national strategy for the northeast region and passed the Black Land Protection Law of the People's Republic of China in 2022 to safeguard cultivated land and ensure food security. The Chinese government has directed its focus towards Northeast China to ensure national food security, with Heilongjiang, Jilin, and Liaoning provinces

outlining detailed plans for grain production. Thus, it is crucial to examine the temporal and spatial dynamics of cultivated land expansion in Northeast China and its potential effects on food security.

Since China's reform and opening (1978–present), China's cultivated land resources have been scarce [7–9]. Academic scholars have conducted a systematic investigation into the spatial–temporal patterns [10], fluctuations in quantity, and influential factors related to the dynamic changes to cultivated land resources [11–13]. Earlier research has identified that the evolutionary trajectory of the spatial–temporal patterns of cultivated land holds significant implications for the effective utilization and allocation of cultivated land resources [10,14], as well as food production, while also interacting with policies aimed at facilitating sustainable development at the regional level [15]. Some studies have investigated how the spatial pattern of cultivated land changes in response to various factors such as climatic conditions, technological advancements, and policy institutions [16,17]. Climate warming can alter the temperature and precipitation conditions required for crop growth, shifting the planting boundary northward and necessitating adjustments to the crop planting structure [18]. Technological advancements have also played a significant role in changing the mode of production from manual labor to mechanization, leading to crop variety optimization and the overcoming of obstacle factors [19,20] Additionally, changes in policy systems have influenced the adjustment of agricultural structure and facilitated the dynamic balance of total cultivated land [21].

Limited research has been conducted to investigate the spatiotemporal dynamics of newly added cultivated land in Northeast China. Notably, certain regions in Northeast China exhibit concentrated fluctuations in cultivated land and conversions between dryland and paddy fields, which have been investigated in prior studies [18,22]. Limited research has been conducted on the vulnerability of recently added cultivated land, with most studies focusing on the overall vulnerability of a given region. To investigate the spatial–temporal pattern of cultivated land, scholars have primarily utilized methods such as the land use transfer matrix (LUTM) [23], remote sensing analysis and simulation [24], and GIS spatial statistics [14,25]. In the realm of geographic research, the SRP (ecological stress, sensitivity, and response) and PSR (stress-state-response) models have been commonly employed to investigate ecological vulnerability. This approach has been utilized by scholars such as Manfré in their studies [26]. Few studies have focused on the spatial–temporal pattern, quality characteristics, and dynamics of newly increased cultivated land and their impact on food security in Northeast China. The research hotspots have been concentrated in the Huang-Huai-Hai Plain [12,27]. This paper analyzes the spatial and temporal pattern and vulnerability of newly increased cultivated land, and its possible impact on grain production in Northeast China during 2000–2020 based on previous studies.

This paper employed remote sensing, temperature and precipitation, and socio-economic data from 2000 to 2020 to construct a cultivated land model to reveal the spatial–temporal pattern of cultivated land change and analyze the marginal effect of newly increased cultivated land and its impact on food security in Northeast China. The paper comprises three sections: revealing the spatial–temporal pattern of cultivated land from 2000 to 2020, constructing a cultivated land vulnerability evaluation system to assess the vulnerability of newly increased cultivated land, and constructing an effect model to analyze the impact of dynamic changes in newly increased cultivated land on grain yield. The study explores the marginal effect and proposes specific measures for ensuring national food security and protecting black land. At the same time, it provides a new way of thinking for the evaluation of cultivated land comprehensive quality, considering the marginal effect of cultivated land in the evaluation of cultivated land comprehensive quality, improving the cultivated land quality evaluation system.

## 2. Materials and Methods

### 2.1. Study Area

Northeast China (115–135° E, 38–56° N) includes Liaoning Province, Jilin Province, Heilongjiang Province, and Chifeng, and Tongliao, Xing'an Meng, and Hulunbuir in the eastern part of the Inner Mongolia Autonomous Region (Figure 1). The topography of Northeast China [28] is semi-annular and triple-banded, with the Daxinganling Mountains to the west, the Xiaoxinganling Mountains to the northwest, and the Changbai Mountains to the east, and the vast northeastern plains within the mountainous hills. Northeast China has fertile and loose black soil, high organic matter content, and an excellent stratified structure, which is ideal for agricultural production. Northeast China is the most extensive commercial grain base in China and is of great importance to ensuring China's food security. Its grain production was 17,346.88 tons in 2020, accounting for 25.91% of the total national grain production, and about 1/3 of the grain is transferred out.

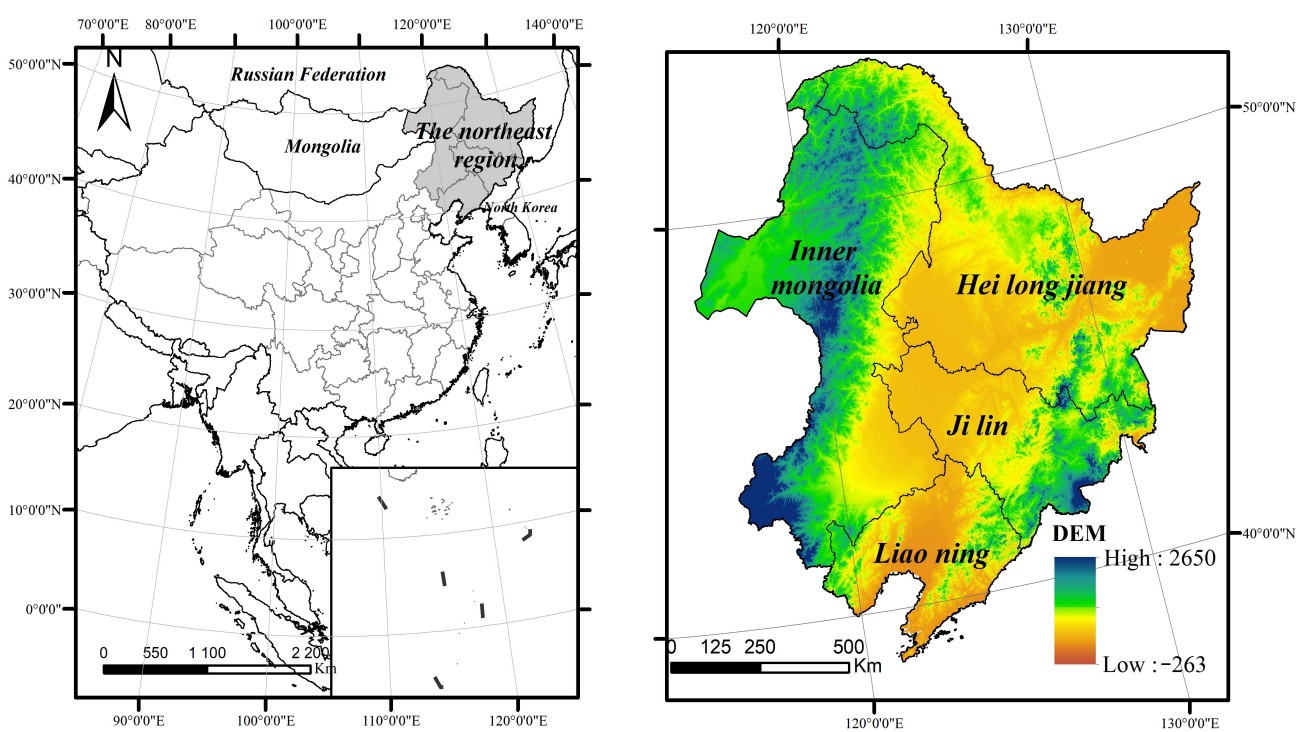

**Figure 1.** The overview of the study area.

### 2.2. Data Source and Preprocessing

The data used in this study include cultivated land data, climate data, soil erosion data, and cultivated land quality data. Cultivated land data were from the remote sensing interpretation of land use in the Resource and Environment Science Data Center of the Chinese Academy of Sciences (http://www.resdc.cn/Datalist1.aspx, accessed on 6 May 2022), which includes 6 categories and 25 subcategories, of which cultivated land is divided into two subcategories, dryland and paddy land (Table 1). Monthly average precipitation and temperature data from 2000 to 2020 were obtained from the China 1 km monthly precipitation and temperature dataset of the National Tibetan Plateau Science Data Center (http://data.tpdc.ac.cn/zh-hans/, accessed on 6 May 2022). Soil erosion data were derived from spatial raster data in the Resource and Environment Science Data Center of the Chinese Academy of Sciences (http://www.resdc.cn/Datalist1.aspx, accessed on 6 May 2022), which includes 3 categories and 16 subcategories. The soil erosion type in Northeast China mainly includes 3 categories and 12 subcategories. Cultivated land quality data sources were from the quality evaluation data of Liaoning, Jilin, and Heilongjiang Provinces and Inner Mongolia. The higher the grade, the worse the quality of cultivated land.

**Table 1.** Land-Use and Land-Cover Change classification table.

| First Class Type | | Secondary Type | | Meaning |
|---|---|---|---|---|
| Numbering | Name | Numbering | Name | |
| 1 | Cultivated land | | | Land for planting crops, including cultivated land that has been in use, newly opened cultivated land, leisure land, land for crop rotation, and land for grass field rotation; agricultural fruit, mulberry, and agricultural and forestry land mainly used for planting crops; beaches and tidal flats. |
| | | 11 | Paddy field | There are related facilities for water source guarantee and irrigation, generally irrigated cultivated land, cultivated land for growing aquatic crops such as rice and lotus root, including cultivated land where rice and dryland crops are planted in turn |
| | | 12 | Dry land | There are no irrigation water sources and facilities, and the cultivated land for growing aquatic crops depends on natural precipitation; the dry crop cultivated land that has a water source and irrigation facilities and can be irrigated normally under normal conditions; the cultivated land mainly for vegetable cultivation; the idle land for crop rotation planting |
| 2 | Woodland | | | Growing trees, shrubs, bamboos, and forestry land such as coastal mangroves |
| 3 | Grassland | | | All kinds of grasslands with a coverage of more than 5% mainly of growing herbs, including nomadic shrub grasslands and sparse forest grasslands with a canopy closure of less than 10% |
| 4 | Waters | | | Natural land waters and land for water conservancy facilities |
| 5 | Urban and rural construction land | | | Urban and rural residential areas and other lands for industry, mining, transportation, etc. |
| | | 51 | Urban land | Large, medium, and small cities and built-up areas above counties and towns |
| | | 52 | Rural settlement | Rural settlements independent of towns |
| | | 53 | Other construction land | Refers to factories and mines, large industrial areas, and other land and traffic roads, airports, and special land. |
| 6 | Unused land | | | Unused land, including difficult-to-use land. |

Cultivated land, temperature, and precipitation data in Northeast China are clipped based on vector boundaries and analyzed comprehensively for the period 2000–2020. The spatial distribution of cultivated land is determined using ArcGIS by correlating and overlaying night-time light, soil erosion type data, and cultivated land quality data. The vulnerability of newly cultivated land is evaluated using the patch shape index of cultivated land, data weights of various indicators are calculated using SPSS, and the relationship between dynamic changes in cultivated land and grain yield is analyzed.

### 2.3. Methodology

#### 2.3.1. Dynamic Change of Cultivated Land

The land use conversion matrix (LUTM) comes from the quantitative description of system states and state transfers in system analysis [23]. The rows and columns of the LUTM (Table 2) represent the land use types at time points $T_1$ and $T_2$. $P_{ij}$ represents the percentage of total land area converted from land type i to land type j during $T_1$–$T_2$; $P_{ii}$ represents the percentage of the area where land use type i remains constant during $T_1$–$T_2$. $P_{i+}$ represents the percentage of the total area of land type i at time point $T_1$. $P_{+j}$ represents the percentage of the total area of land use type j at time point $T_2$.

**Table 2.** Land use transfer matrix [23].

| | | T2 | | | | $P_{+i}$ | Reduce Area |
|---|---|---|---|---|---|---|---|
| | | $A_1$ | $A_2$ | ... | $A_n$ | | |
| | $A_1$ | $P_{11}$ | $P_{12}$ | ... | $P_{1n}$ | $P_{1+}$ | $P_{1+}$–$P_{11}$ |
| | $A_2$ | $P_{21}$ | $P_{22}$ | ... | $P_{2n}$ | $P_{2+}$ | $P_{2+}$–$P_{22}$ |
| T1 | ... | ... | ... | ... | ... | ... | ... |
| | $A_n$ | $P_{n1}$ | $P_{n2}$ | ... | $P_{nn}$ | $P_{n+}$ | $P_{n+}$–$P_{nn}$ |
| | $P_{+j}$ | $P_{+1}$ | $P_{+2}$ | ... | $P_{+n}$ | 1 | |
| | Add area | $P_{+1}$–$P_{11}$ | $P_{+2}$–$P_{22}$ | ... | $P_{+n}$–$P_{nn}$ | | |

#### 2.3.2. VSD Model Analysis

A hierarchical analysis-based Exposure–Sensitivity–Responsiveness (VSD) model was employed to calculate the vulnerability index of newly increased cultivated land, where the degree of exposure was determined by natural and socio-economic characteristics (Figure 2). Changes in population and land use patterns were used to reflect the degree of exposure, where a higher degree of exposure was found to increase sensitivity to ecological and environmental risks, resulting in higher vulnerability [17,28]. The sensitivity degree of the newly increased cultivated land is determined by the potential for ecological problems or threats, including soil erosion, heat conditions, and other natural factors. Higher sensitivity indicates a greater likelihood of damage. The responsiveness degree is determined by the ability of the cultivated land to resist external disturbances or stress, which can be affected by human intervention or adaptive management practices [29,30].

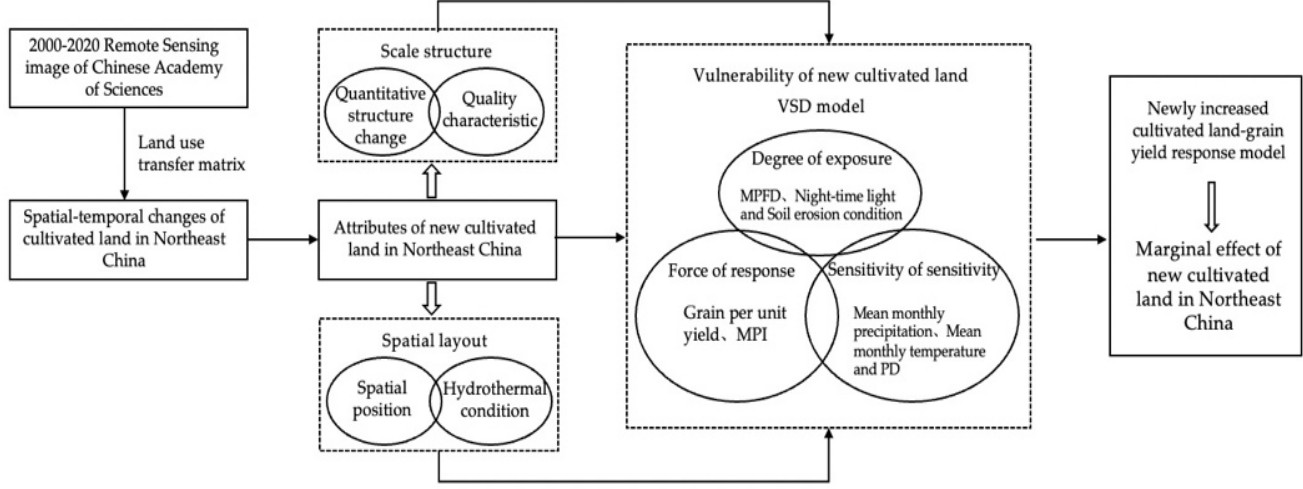

**Figure 2.** Study flow chart.

This study selected the average surface integral shape dimension and night-time light to represent the exposure sign of newly increased cultivated land. The sensitivity was characterized by changes in soil erosion, water and heat conditions, and patch density of

the newly increased cultivated land. The responsiveness was characterized by the patch dominance and grain yield of the newly increased cultivated land [26].

(1)  Analytic hierarchy process

Hierarchical analysis is a combined qualitative and quantitative decision analysis method for multi-objective complex problems and is widely used in various types of research [25,31], for example, cultivated land comprehensive quality assessment, village development assessment, etc. This study constructs an evaluation system based on hierarchical analysis to evaluate the ecological vulnerability characteristics of newly increased cultivated land [32].

$$A = (X_{ij})_{n \times n} = \begin{pmatrix} x_{11} & \cdots & x_{1n} \\ \vdots & \ddots & \vdots \\ x_{n1} & \cdots & x_{nn} \end{pmatrix} W_i = \frac{\overline{W}_i}{\sum_{i=1}^{n} \overline{W}_i}, \ W = \begin{pmatrix} W_1 \\ W_n \end{pmatrix} CI = \frac{\lambda_{max} - n}{n - 1} CR = CI/RI$$

where A is the orthogonal matrix, $X_{ij}$ is the comparison result of the i relative to the j, $W_{ij}$ is the weight and eigenvector of each evaluation index, CI is the consistency test of the index, the $\lambda$ is the maximum characteristic root, and RI is the random consistency index.

(2)  Landscape pattern indices

    a.     Density of patches (PD)

$$PD = n_i / A(100)$$

where $n_i$ is the total area of landscape elements of category i; A is the total area of all landscapes [14].

    b.     Average plaque typing dimension (MPFD)

$$MPFD = \sum_{i=1}^{m} \sum_{j=1}^{n} \left( \frac{2Ln\left(0.25p_{ij}\right)}{Lna_{ij}} \right) \frac{a_{ij}}{A}$$

where m is the number of patch types, n is the number of patches of a certain type, $p_{ij}$ is the perimeter of patch ij, $a_{ij}$ is the area of patch ij, and A is the area of the total landscape [27].

    c.     Largest Patch Index (LPI)

$$LPI = \frac{1}{-\sum_{i=1}^{m}(P_i \ln P_i)}$$

where: $P_i$ is the proportion of landscape patches type i.

(3)  Vulnerability index of cultivated land ecosystem

$$V = \sum_{i=1}^{n} X_i \times W_i$$

where V denotes the cropland ecosystem vulnerability index, $X_i$ is the data of each indicator, and Wi is the weight of each indicator data.

2.3.3. Newly Increased Cultivated Land–Grain Yield Response Model

As the concept of sustainable development evolves, cultivated land security now encompasses quantity, quality, and ecological aspects (Figure 2). Thus, ensuring the trinity of cultivated land security is essential for stable long-term food production [24,30]. Studies examining the relationship between cultivated land quality, ecological security, ecosystem vulnerability, and food production are insufficient [8]. Cultivated land quality is typically assessed by its quality class, while the impact of ecological security on food production is evaluated [18]. This study examines the relationship between the comprehensive change of

newly increased cultivated land and grain yield. The characterization scale is the area of newly increased cultivated land, the characterization quality is soil erosion, and the degree of vulnerability is characterized by the vulnerability, temperature, and precipitation of cultivated land.

$$r = \frac{l_{xy}}{\sqrt{l_{xx}l_{xy}}} = \frac{\sum_{i=1}^{n} \frac{(x - \bar{x})(y - \bar{y})}{n-1}}{\sqrt{\frac{\sum_{i=1}^{n}(x - \bar{x})^2}{n-1}}} \times \frac{1}{\sqrt{\frac{\sum_{i=1}^{n}(y - y)^2}{n-1}}}$$

where r is the correlation coefficient, with the range of $[-1, 1]$. n is the number of factors, x is the data of various indicators of cultivated land dynamics, which characterize the scale, quality, and ecological environment of cultivated land, and y is the grain yield. The correlation model was tested using the chi-square test pair (Sig), which indicates a significant correlation between the two factors when the *p*-value is $<0.05$ and a highly significant correlation between the two factors when the *p*-value is $<0.01$ [11].

### 3. Results

*3.1. Spatial–Temporal Pattern Changes of Cultivated Land*

3.1.1. Overall Change in Cultivated Land

According to the land use transfer matrix calculation, between 2000 and 2020, Northeast China experienced a 3.08% increase in cultivated land (Table 3), adding 1,778,500 hectares, primarily from previously unused land, forests, and grasslands. The increase in paddy land was approximately 1651,800 ha, while the dry land decreased by about 4844 ha. Notably, 2,122,000 hectares of dry land were converted into paddy fields, accounting for 58.27% of the increase in paddy fields (Figure 3). The newly occupied cultivated land was mainly used for afforestation or reclamation and urban development. In summary, the RSI values are larger in the central part of the plain and smaller in the peripheral areas; the RSI values in the peripheral areas of the cities are significantly higher than those in the other areas, and the area of rural settlements increased significantly due to urban radiation. The topography and proximity to the city may influence the evolution of rural settlements.

**Table 3.** Land use type area conversion table from 2000 to 2020 (ha).

| 2000 \ 2020 | 2 | 3 | 4 | 6 | 11 | 12 | 51 | 52 | 53 |
|---|---|---|---|---|---|---|---|---|---|
| 2 | 453,846.58 | 10,782.58 | 1735.07 | 11,496.98 | 2596.69 | 19,953.17 | 264.06 | 813.61 | 423.65 |
| 3 | 15,174.52 | 203,016.43 | 677.82 | 12,143.82 | 1410.18 | 9189.46 | 129.33 | 440.62 | 417.94 |
| 4 | 777.90 | 598.32 | 20,044.00 | 5331.56 | 1378.22 | 2828.83 | 110.54 | 100.92 | 234.41 |
| 6 | 2761.97 | 3331.18 | 1614.55 | 51,494.15 | 4504.95 | 6626.86 | 88.28 | 197.41 | 162.89 |
| 9 | 0.00 | 0.00 | 2.20 | 0.00 | 0.00 | 0.00 | 0.00 | 0.00 | 2.04 |
| 11 | 670.67 | 348.12 | 582.44 | 603.05 | 29,761.13 | 11,530.90 | 390.95 | 974.09 | 227.22 |
| 12 | 16,393.51 | 6163.04 | 2619.18 | 3502.85 | 21,219.98 | 265,911.95 | 2144.36 | 6665.36 | 1131.02 |
| 51 | 28.86 | 12.22 | 17.34 | 5.56 | 22.10 | 177.28 | 4169.26 | 167.12 | 23.43 |
| 52 | 451.74 | 240.52 | 119.90 | 147.43 | 806.01 | 4632.80 | 931.96 | 15,659.76 | 161.85 |
| 53 | 47.31 | 30.97 | 471.94 | 23.91 | 17.42 | 56.46 | 237.68 | 30.60 | 553.52 |

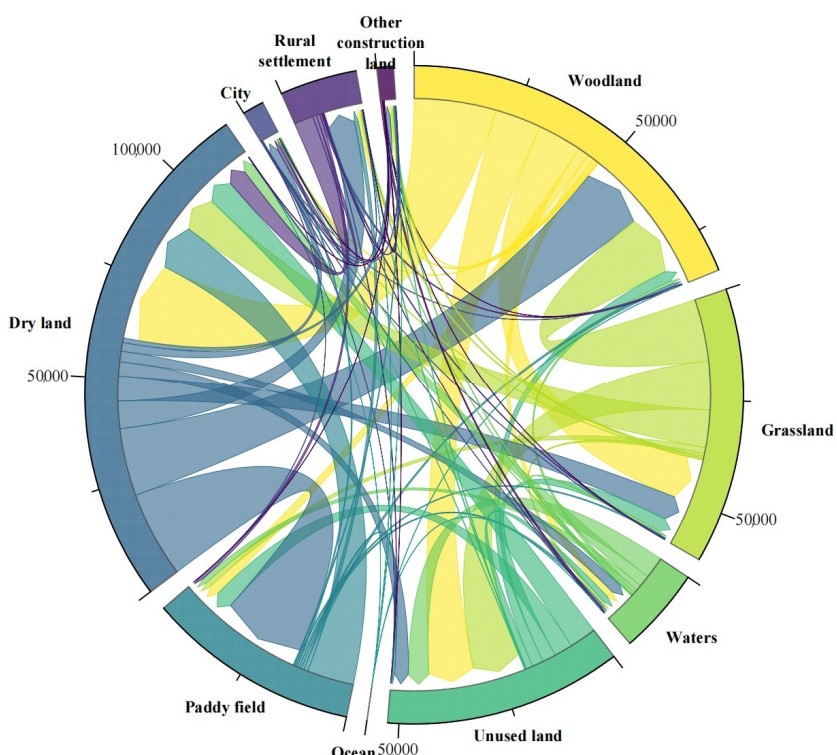

**Figure 3.** Land type transfer table.

3.1.2. Spatial–Temporal Pattern of Newly Increased Cultivated Land

The spatial distribution of newly increased cultivated land was concentrated in the Northeast Plain and the area between the Lesser Khingan Mountains and the Sanjiang Plain, as well as in the northwest and northeast of Heilongjiang Province, the south of the Eastern Fourth Lian of Inner Mongolia, and the southwest of Liaoning Province (Figure 4). Additionally, in the northeast Sanjiang Plain, there was a phenomenon of returning dry land to cultivated land.

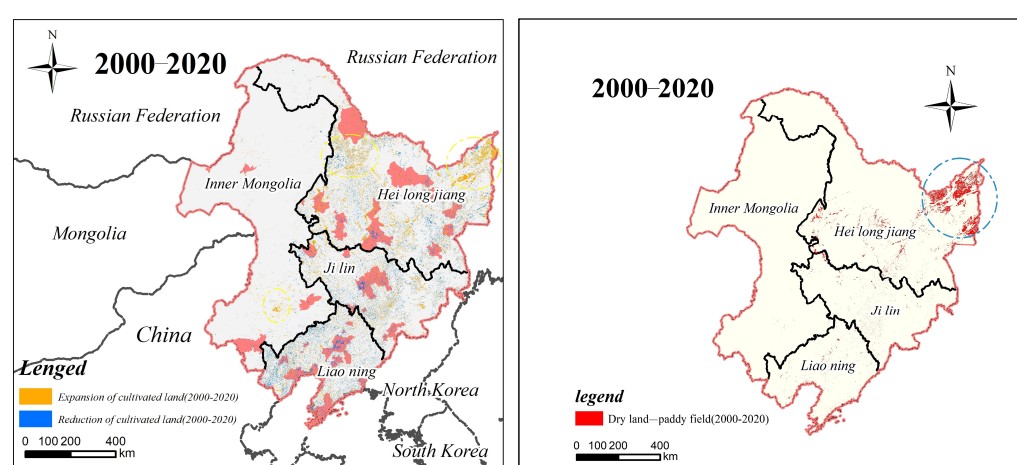

**Figure 4.** Changes of cultivated spatial land pattern.

In Northeast China, temperature higher generally leads to planting boundary expansion to the north [33]. The newly increased cultivated land due to warming is mainly located in the temperature-limited northern region, where extreme climate conditions may lead to decreased or no crop yield (Figure 5). Over the period 2000 to 2020, Northeast China experienced significant precipitation reduction, with 85% of the region having an average monthly rainfall of less than 5 mm, thereby increasing the likelihood of drought.

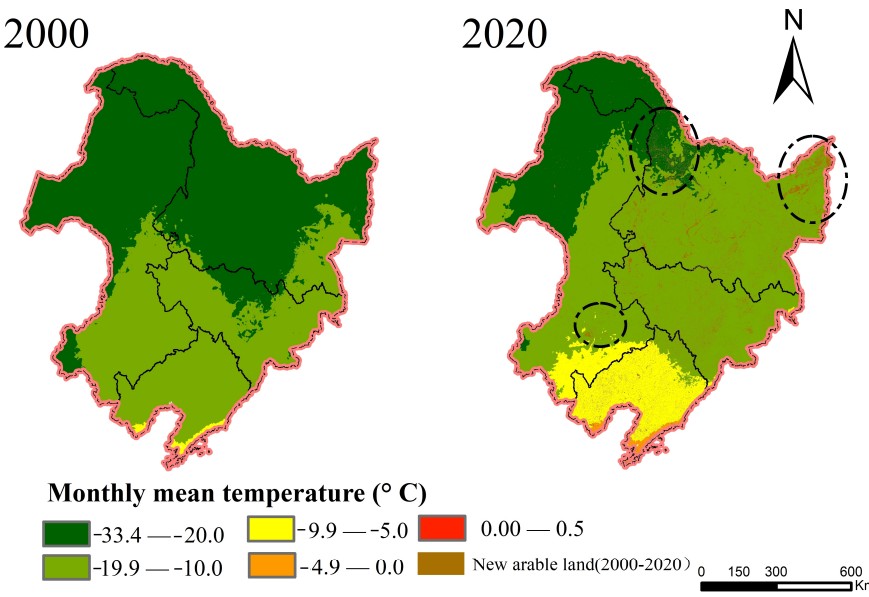

**Figure 5.** Monthly mean temperature pattern changes.

With the change in water and heat conditions, the cultivated land in Northeast China expanded northward, but most newly increased cultivated land is in low-temperature and arid regions (Figure 6). The yield of cultivated land is significantly affected by temperature fluctuations and stabilizing the yield of newly increased cultivated land is challenging.

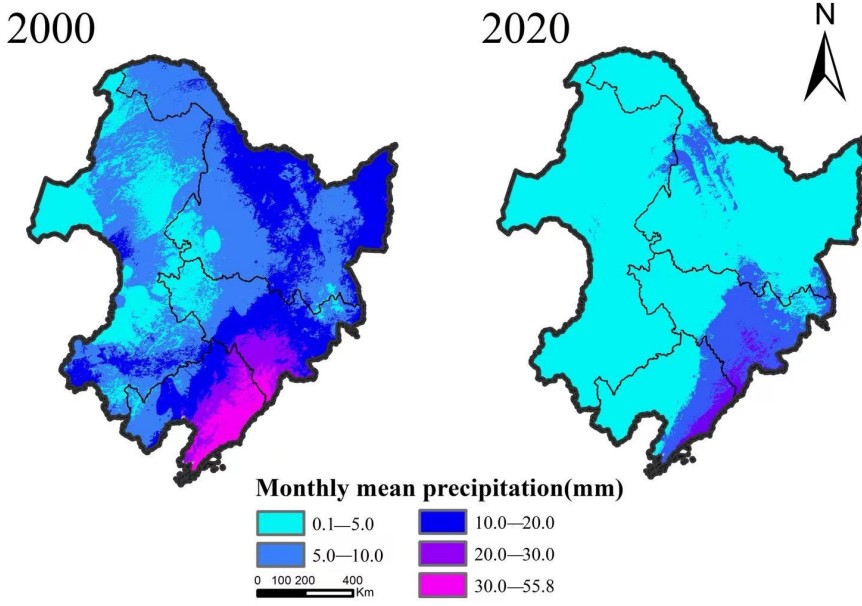

**Figure 6.** Monthly mean precipitation changes.

### 3.1.3. Newly Cultivated Land Quality Attributes

The quality of cultivated land in Northeast China ranged from grade 1 to grade 10 from 2000 to 2020, with the Northeast Plain area having the highest concentration of cultivated land (Figure 7), and mountainous and hilly areas having low-quality cultivated land. Most of the newly increased cultivated land in Northeast China falls within grades 6–10.

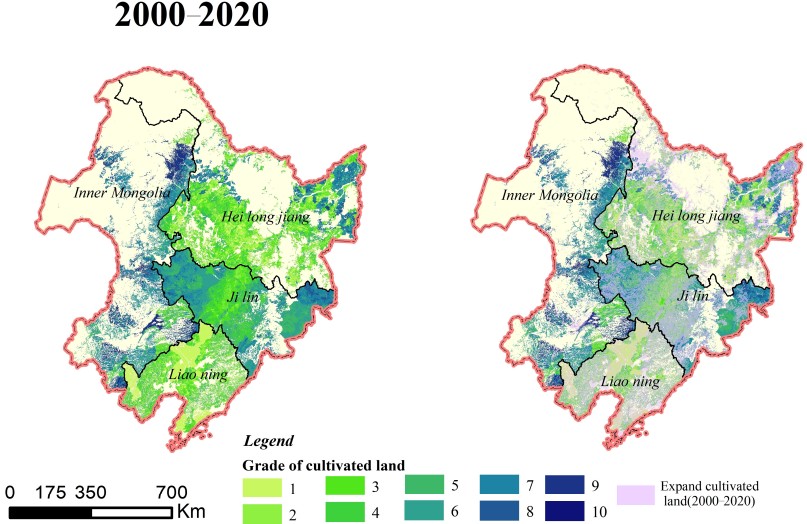

**Figure 7.** Grade map of cultivated land in Northeast China.

The newly increased cultivated land in Northeast China comprises mainly quality grade 6–10, with 58.54% falling in this range (Figure 8). Within this range, the cultivated land with quality grade 8–10 accounts for 35.58%, and the cultivated land with quality grade 10 accounts for 11.89%. In contrast, the reduced cultivated land is grade 1 high-quality cultivated land. Although the quality of newly increased cultivated land in Northeast China is low, the occupied cultivated land is of high quality.

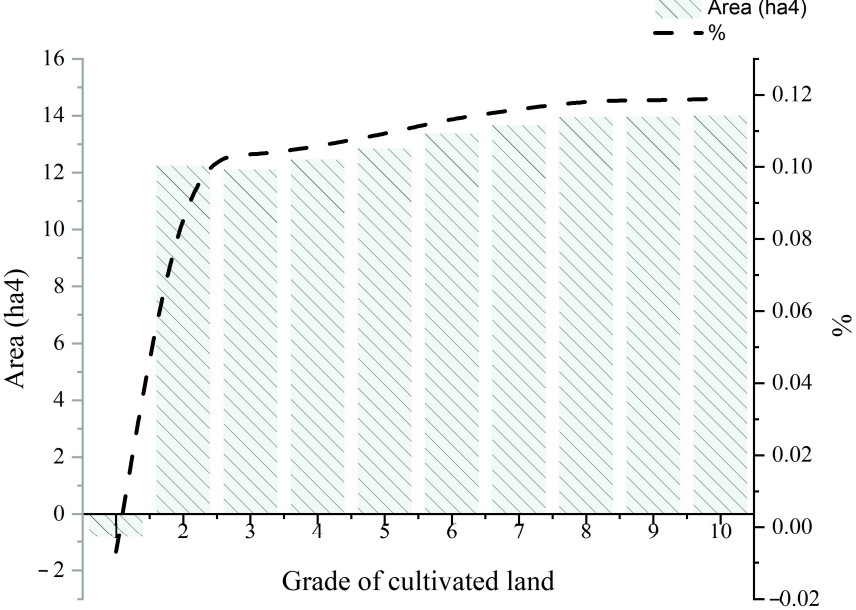

**Figure 8.** Status of cultivated land expansion in Northeast China.

### 3.2. Vulnerability of New Cultivated Land

In Northeast China, moderately vulnerable areas of newly increased cultivated land were mainly distributed in the eastern and southwestern regions (740,600 hectares, 62.84%), while mildly vulnerable areas were primarily found in the Northeast Plain and northern and central areas of the Greater Khingan Mountains (395,100 hectares, 33.53%). Severely vulnerable cultivated land was mainly located in the northern part of the Liaodong Peninsula (4.28 million hectares, 3.6%). The proportion of newly increased cultivated land in medium–highly vulnerable areas was twice as high (66.44%) as that in mildly vulnerable

areas. The concentration of newly increased cultivated land in Northeast China is in commercial grain production bases, such as the Sanjiang Plain and eastern Northeast Plain. Thus, optimizing the ecological security of the newly increased cultivated land is necessary to ensure its quality and safety and promote stable grain production. The following results demonstrate this (Tables 4–6 and Figure 9).

**Table 4.** Index judgment matrix.

| Index Data | MPFD | Index of Light | Soil Erosion Condition | PD | Mean Monthly Precipitation | Mean Monthly Temperature | LPI | Grain per Unit Yield |
|---|---|---|---|---|---|---|---|---|
| MPFD | 1.00 | 0.50 | 0.67 | 1.00 | 1.25 | 1.25 | 1.00 | 0.50 |
| Index of light | 2.00 | 1.00 | 1.25 | 1.43 | 1.67 | 1.67 | 1.43 | 1.11 |
| Soil erosion condition | 1.50 | 0.80 | 1.00 | 1.11 | 0.83 | 0.83 | 5.00 | 0.50 |
| PD | 1.00 | 0.70 | 0.90 | 1.00 | 0.83 | 0.83 | 1.00 | 0.50 |
| Mean monthly precipitation | 0.80 | 0.60 | 1.20 | 1.20 | 1.00 | 1.00 | 0.77 | 0.50 |
| Mean monthly temperature | 0.80 | 0.60 | 1.20 | 1.20 | 1.00 | 1.00 | 0.50 | 0.40 |
| LPI | 1.00 | 0.70 | 0.20 | 1.00 | 1.30 | 2.00 | 1.00 | 0.56 |
| Grain per unit yield | 2.00 | 0.90 | 2.00 | 2.00 | 2.00 | 2.50 | 1.80 | 1.00 |

**Table 5.** Random consistency check form.

| Random Consistency Table | | | | | | | | | | | | | | |
|---|---|---|---|---|---|---|---|---|---|---|---|---|---|---|
| n | 3 | 4 | 5 | 6 | 7 | 8 | 9 | 10 | 11 | 12 | 13 | 14 | 15 | 16 |
| RI | 0.52 | 0.89 | 1.12 | 1.26 | 1.36 | 1.41 | 1.46 | 1.49 | 1.52 | 1.54 | 1.56 | 1.58 | 1.59 | 1.59 |
| n | 17 | 18 | 19 | 20 | 21 | 22 | 23 | 24 | 25 | 26 | 27 | 28 | 29 | 30 |
| RI | 1.6 | 1.61 | 1.62 | 1.63 | 1.64 | 1.64 | 1.65 | 1.65 | 1.66 | 1.66 | 1.66 | 1.67 | 1.67 | 1.67 |

**Table 6.** The index weight of cultivated land ecosystem vulnerability.

| Layer of Criterion | Index Data | Index of Weight |
|---|---|---|
| Degree of exposure | Mean fractal dimension of plaque (MPFD) | 0.10 |
| | Night-time light | 0.17 |
| | Soil erosion condition | 0.13 |
| | Density of patches (PD) | 0.10 |
| Degree of sensitivity | Mean monthly precipitation | 0.10 |
| | Mean monthly temperature | 0.09 |
| Force of response | Maximum plaque index (MPI) | 0.10 |
| | Grain per unit yield | 0.20 |

Heilongjiang Province has the largest area of newly increased cultivated land, followed by Jilin Province, Liaoning Province, and the Inner Mongolia Autonomous Region. Most of the newly increased cultivated land in Northeast China is moderately vulnerable, except for the four alliances in eastern Inner Mongolia where the mildly vulnerable areas are more significant. Severely vulnerable cultivated land accounted for 21.27% of the newly increased cultivated land in Liaoning Province. Heilongjiang Province has the largest increase in cultivated land (677,000 hectares), but 66.62% of the newly increased cultivated land is in moderately vulnerable areas. Jilin Province increased 245,700 hectares of cultivated land, of which 63.13% was in moderately vulnerable areas. Liaoning Province increased 2.03 million hectares of cultivated land, of which 76.68% was in moderately vulnerable areas. In four cities in eastern Inner Mongolia, 54,800 hectares of newly cultivated land were increased, of which 41.97% was in moderately vulnerable areas (Figure 10).

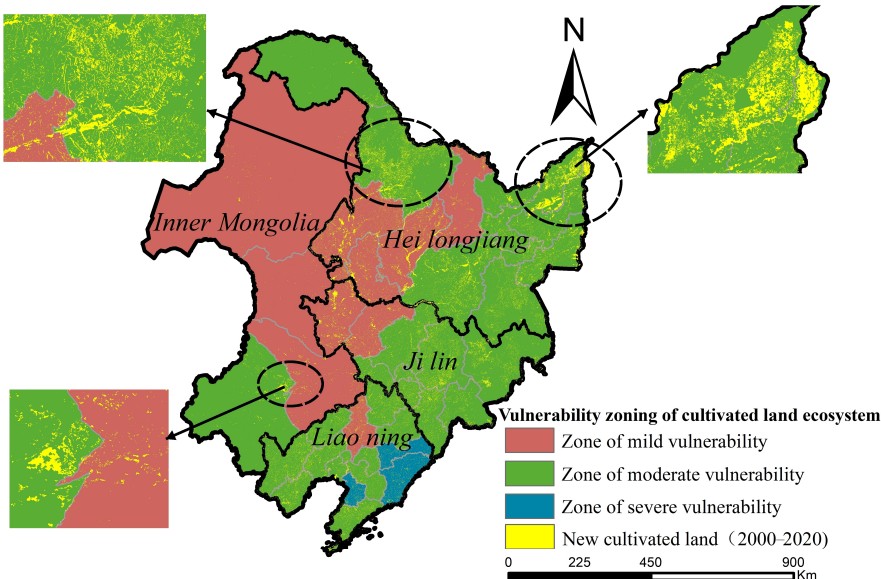

**Figure 9.** Spatial pattern of ecosystem vulnerability in Northeast China.

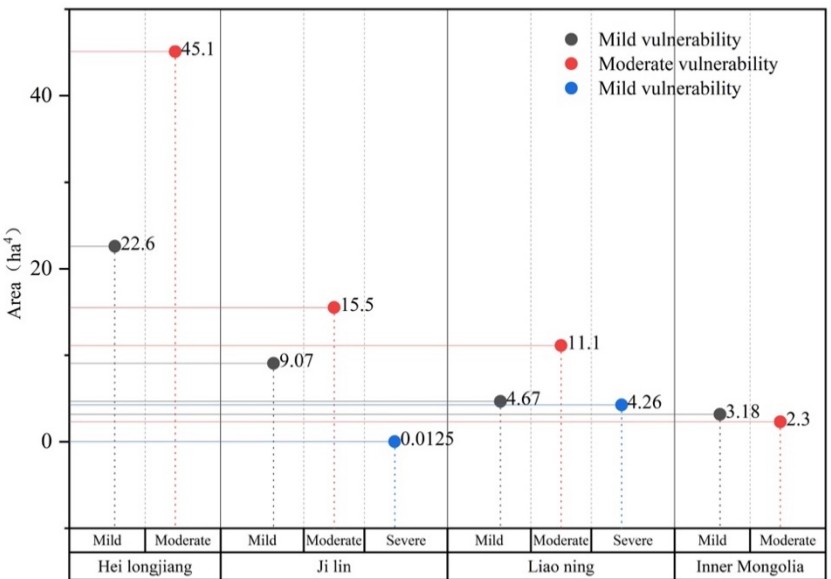

**Figure 10.** Statistics of cultivated land ecosystem vulnerability in Northeast China.

Most of Northeast China's newly increased cultivated land is in moderately vulnerable areas, leading to concerns regarding the quality and ecological security of the land. Unstable grain production and output from this land may impact overall grain production and potentially pose a threat to food security.

### 3.3. Marginal Effect of New Arable Land on Grain Yield

In this study, the correlation effect model was used to analyze the relationship between the characteristics of changes in the quantity, quality, and ecological dynamics of cultivated land and grain yield.

Table 7 shows a significant negative correlation between grain yield and vulnerability of newly increased cultivated land, indicating that higher vulnerability of cultivated land leads to higher grain yield and lower regional ecosystem resilience to external interference, which affects the material cycle or energy flow of the cultivated land ecosystem [34]. Higher grain yields in Northeast China are associated with greater ecological environment costs,

as demonstrated by a significant positive correlation between grain yield and temperature change, while light conditions remain an essential factor affecting grain yield.

**Table 7.** The relationship between landscape indicators and variables in rural settlements.

| Analysis of Correlation | | New Area of Cultivated Land | Ecological Vulnerability | Quality of Cultivated Land | Variation of Temperature | Variation of Precipitation | Production of Grain |
|---|---|---|---|---|---|---|---|
| New area of cultivated land | Pearson correlation | 1.00 | 0.06 | −0.16 | −0.373 * | −0.03 | −0.25 |
| | Sig | | 0.73 | 0.32 | 0.02 | 0.84 | 0.12 |
| Ecological vulnerability | Pearson correlation | 0.06 | 1.00 | 0.10 | −0.417 ** | −0.481 ** | −0.335 * |
| | Sig. | 0.73 | | 0.54 | 0.01 | 0.00 | 0.03 |
| Quality of cultivated land | Pearson correlation | −0.16 | 0.10 | 1.00 | 0.24 | −0.477 ** | 0.05 |
| | Sig. | 0.32 | 0.54 | | 0.14 | 0.00 | 0.75 |
| Variation of temperature | Pearson correlation | −0.373 * | −0.417 ** | 0.24 | 1.00 | 0.15 | 0.511 ** |
| | Sig. | 0.02 | 0.01 | 0.14 | | 0.37 | 0.00 |
| Variation of precipitation | Pearson correlation | −0.03 | −0.481 ** | −0.477 ** | 0.15 | 1.00 | 0.11 |
| | Sig. | 0.84 | 0.00 | 0.00 | 0.37 | | 0.48 |
| Production of grain | Pearson correlation | −0.25 | −0.335 * | 0.05 | 0.511 ** | 0.11 | 1.00 |
| | Sig. | 0.12 | 0.03 | 0.75 | 0.00 | 0.48 | |

** At level 0.01 (two-tailed), the correlation was significant. * At level 0.05 (two-tailed), the correlation was significant.

The expansion of cultivated land area is significantly negatively correlated with regional temperature, indicating that temperature stability is a crucial factor affecting cultivated land expansion. The newly increased cultivated land mainly comes from ecological land types such as woodland, grassland, and wetlands, which will inevitably undermine the original ecosystem. The dynamic changes in the spatial and temporal patterns of cultivated land may pose a risk of food production instability. However, large-scale cultivated land reclamation beyond the self-restoration ability of the ecosystem balance can result in severe consequences such as ecosystem disruption or collapse.

## 4. Discussion

### 4.1. Newly Increased Cultivated Land Is Unstable Due to External Disturbance

With increasing urbanization, urban expansion will occupy more agricultural land, which is irreversible [14]. The inadequacies of China's cultivated land occupation and compensation balance system are evident, as the replenished cultivated land is of lower quality than the high-quality cultivated land that has been occupied, leading to a significant gap in the quality of cultivated land compensation [15].

Most of the newly increased cultivated land is situated in the northern part of Northeast China, primarily in the terrain fluctuation area where the Sanjiang Plain, the Greater Khingan Mountains, and the Lesser Khingan Mountains converge. Among them, Sanjiang Plain's new arable land area is the largest. Sanjiang Plain, a biodiverse wetland area, is a significant increase area of newly cultivated land in Northeast China from 2000 to 2020 [26]. Most of the newly increased cultivated land in the Sanjiang Plain is converted from wetland, woodland, or grassland [35]. Disturbance beyond ecosystem recovery capacity may cause severe ecological problems, risking soil erosion, increased land degradation, and lower cultivated land stability. Global warming has resulted in a significant increase in the average monthly temperature, causing the northward shift of the crop cultivation boundary [21,36].

From 2000 to 2020, the average monthly precipitation in Northeast China decreased significantly, and the drought trend was evident. The uncertainty of climate conditions



may induce flood and drought disasters and affect crop yield [37]. This region experiences an average monthly temperature ranging from −10 °C to −20 °C and an average monthly precipitation of 0.15 mm and 10 mm. The average monthly temperature of the newly increased cultivated land is −10~−20 °C and the average monthly precipitation is 0.15~10 mm. Low temperature, limited precipitation, and unstable hydrothermal conditions may lead to uncertainty and instability of newly increased cultivated land yield [38]. The cultivated land in Northeast China lacks quality and ecological security, exhibits weak stability, and has a low ability to resist external interference. The poor stability of newly increased cultivated land in Northeast China reflects the defects in China's cultivated land occupation and compensation balance system, which cannot guarantee the comprehensive quality of cultivated land and only emphasizes the quantity balance of cultivated land, ignoring the quality of cultivated land and ecological stability. It suggests that scholars should deepen the research on the quality security and ecological security of the newly increased cultivated land to comprehensively measure the newly increased cultivated land and ensure its stability.

### 4.2. Newly Increased Cultivated Land Vulnerability Poses a Potential Threat to Food Security

Northeast China belongs to the temperate monsoon continental climate, making summer precipitation more and more concentrated and soil erosion more serious. Secondly, the central hinterland of Northeast China is plain, lacking protective forest network obstruction, and some land has wind erosion hazards. Soil and water loss is the main reason for black soil thinning, presenting an urgent problem to be addressed [17]. From 2000 to 2020, 91.61% (1.07 million hectares) of newly increased cultivated land in Northeast China was at risk of water erosion, mainly in the Sanjiang Plain and the confluence of the Greater Khingan Mountains and Lesser Khingan Mountains. In total, 8.11% (95,800 hectares) of cultivated land is at risk of wind erosion, mainly distributed in the arid and semi-arid regions of Inner Mongolia. In addition, about 63.13% of the newly increased cultivated land in Northeast China has medium to high vulnerability, with short-term potential to increase production but long-term risks.

The fragility of the ecological system and uncertainty of climatic conditions raise the risk of the collapse of the cultivated land production system, resulting in reduced crop yields or even complete crop failure, posing a significant threat to food security [39]. The ecological stability of cultivated land can enhance its self-regulation and self-recovery capabilities, thereby preventing or mitigating ecological problems and disasters [16]. Ensuring the quality and ecological security of newly increased cultivated land and improving the resistance and resilience ability of the cultivated land ecosystem to external interference is essential to ensure the stability of the cultivated land production system, increase grain output growth, and guarantee food security [6].

### 4.3. Proposals for Optimizing Current Cultivated Land Protection Policies

Despite the Chinese government's implementation of strict policies and systems, such as the cultivated land occupation and compensation balance system and the designation of permanent basic cultivated land protection areas, some issues with the protection of cultivated land persist [28]. Song suggested prioritizing the development of high-quality cultivated land to ensure both quantity and quality protection, as the current cultivated land balance system focuses on quantity and may negatively affect crop yield and national food security (Song and Pijanowski 2014). The permanent basic cultivated land construction overlooks the comprehensive development of the cultivated land ecosystem and does not align with the Chinese government's agricultural modernization goals, emphasizing only rural roads and other supporting facilities [40]. Since China's cultivated land has reached a saturation point, it is challenging to increase its quantity further [8,18]. Thus, the focus should shift from quantity protection to quality and ecological protection, and a stable cultivated land output mechanism must be established for the long term [15].

China is experiencing a decline in cultivated land quality, and non-grain cultivated land is becoming increasingly prevalent [13]. To ensure national food security, the Chinese government has developed policies and measures for major grain-producing areas [5]. In 2022, the Chinese government elevated the black land protection strategy in major grain-producing regions of Northeast China to a national strategy. This was achieved through the implementation of strict policies, laws, and regulations aimed at strengthening the protection of black land. The government also encouraged individual farmers to actively practice black land protection measures, establish a long-term and stable protection mechanism, and ensure the sustainable development of cultivated land for food production stability [16]. To ensure the stability of the cultivated land production system and increase grain output in Northeast China, the efficiency and quality of cultivated land should be fully considered when implementing the cultivated land protection policy strictly. Additionally, the construction of high-standard cultivated land should be advanced to improve cultivated land ecosystems, enhance their resilience, and ensure the quantity, quality, and ecological security of cultivated land [36]. Promoting large-scale land management by integrating tenure relations and a clustered spatial structure can provide basic conditions for conservation farming measures and promote cultivated land protection practices [39].

The Chinese government is committed to ensuring national food security and has implemented food security in cultivated land, which is closely related to national food security. This paper focuses on the newly increased cultivated land, analyzes the marginal effect of grain production, and explores the impact on food security. This paper innovatively discusses the potential impact of the marginal effect of newly increased cultivated land on food security, providing direction for the government to develop comprehensive measures to protect cultivated land and strengthen the overall quality of cultivated land.

## 5. Conclusions

Focusing on the newly increased cultivated land in Northeast China, this paper analyzes the spatial and temporal pattern of newly increased cultivated land. It assesses the marginal effect of newly increased cultivated land's comprehensive attributes on grain production and its potential impact on food security. The study found that from 2000 to 2020, 1.1785 million hectares of newly increased cultivated land in Northeast China were mainly located in the temperature-restricted area in northern Heilongjiang Province, Sanjiang Plain, and the arid and semi-arid area in southern Dongsi League of Inner Mongolia. Most of the newly increased cultivated land had poor quality, with 58.54% of it being at grade 6–10, and the reduced cultivated land was all at grade 1. Additionally, 62.84% of the newly increased cultivated land was in ecologically fragile areas, while the rest was in mildly and severely vulnerable areas. Temperature instability was negatively correlated with cultivated land expansion, while grain production was negatively correlated with cultivated land vulnerability. The increase in grain production at the expense of cultivated land ecology is a potential threat to national food security.

The poor quality of newly increased cultivated land in Northeast China, characterized by ecological fragility, may lead to short-term gains in grain yield but may not guarantee long-term stability. This study found a significant negative correlation between grain yield and cultivated land ecological vulnerability in Northeast China. Therefore, we should take increasing high quality and good ecological cultivated land as the key measures of cultivated land protection. We will give priority to incorporating ecologically sound cultivated land into high-standard cultivated land development zones and strengthen protection. This can not only save the maintenance cost of cultivated land and improve the production efficiency of cultivated land, but also ensure the long-term stability of grain production to the greatest extent.

**Author Contributions:** Conceptualization, X.W. and J.W.; methodology, J.W.; validation, J.W., G.D. and X.W.; formal analysis, X.W.; data curation, X.W. and H.Z.; writing—original draft preparation, X.W. and Y.L.; writing—review and editing, X.W., G.D. and J.W.; visualization, X.W.; supervision, J.W. All authors have read and agreed to the published version of the manuscript.

**Funding:** This research was supported by the Strategic Priority Research Program of the Chinese Academy of Sciences (Grant No. XDA28130400), the National Natural Science Foundation of China (Grant No. 42171266), the National Key R&D Program of China (Grant No. 2021YFD1500101), and the Youth Talent Project of the Northeast Agricultural University of China (Grant No. 19QC35).

**Data Availability Statement:** Data are available in a publicly accessible repository.

**Acknowledgments:** We would like to thank the Center for Resource and Environmental Science and Data, Chinese Academy of Sciences (http://www.resdc.cn/Datalist1.aspx, accessed on 6 May 2022) for the interpretation of land use remote sensing data and the Nanjing Institute of Soil Sciences, Chinese Academy of Sciences (http://soil.geodata.cn/, accessed on 6 May 2022) of China's 1 km raster soil organic matter data. In addition, we would like to thank anonymous reviewers for their valuable comments and suggestions to improve this paper.

**Conflicts of Interest:** The authors declare no conflict of interest.

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
