# Peer review of "Analysis of the Spatial–Temporal Pattern of the Newly Increased Cultivated Land and Its Vulnerability in Northeast China"

_land, doi:10.3390/land12040796_

Round 1

Reviewer 1 Report

Thank you for this contribution, which presents a very important investigation of land use changes in Northeast China. You have definitely chosen an important topic for your research. However, you need to present your approach and your results much more clearly. Non-experts will not be able to understand your results. You do not refer to equations, tables and figures in the text. Some claims are not adequately referenced. You need to create a more readable flow, add more signposting and explain more thoroughly how you came to your conclusions. Also, we need to know about your contribution to the literature.

See attached file.

Good luck with editing this article.

Author Response

Dear Editors and Reviewers:

Thank you very much for giving us the opportunity to revise our manuscript entitled " Analysis on the spatial-temporal pattern of the newly increased Cultivated land and its vulnerability in Northeast China (land-2300733) ". We thank the editors and reviewers for their valuable time and insightful comments. These comments are very helpful for us to improve the quality of the manuscript. We carefully studied the comments and revised the manuscript one by one based on constructive comments. The main corrections to the paper and the responses to the reviewers' comments are detailed as follows this file.

Reviewer 2 Report

The study appears to be well-designed and based on a rigorous analysis of remote sensing data, temperature and precipitation data, and socio-economic data from 2000 to 2020. The authors employ a cultivated land vulnerability evaluation system to assess the vulnerability of newly increased cultivated land and construct an effect model to analyze the impact of dynamic changes in newly increased cultivated land on grain yield. The findings of this research are important for policymakers and stakeholders who are interested in ensuring compliance with China's cultivated land preservation policy. However, further research is needed to explore the long-term impacts of newly increased cultivated land on food security and environmental sustainability in Northeast China. Overall, this manuscript provides valuable insights into the spatial-temporal pattern of newly increased cultivated land and its vulnerability in Northeast China.

one potential weakness of the study is that it relies heavily on remote sensing data, which may have limitations in accurately capturing ground-level changes in cultivated land. Additionally, the study only focuses on Northeast China and may not be generalizable to other regions or countries. Finally, while the study provides valuable insights into the spatial-temporal pattern of newly increased cultivated land and its vulnerability, it does not explore the underlying drivers of cultivated land expansion or provide policy recommendations for addressing this issue.

Author Response

Dear Editors and Reviewers:

Thank you very much for giving us the opportunity to revise our manuscript entitled " Analysis on the spatial-temporal pattern of the newly increased Cultivated land and its vulnerability in Northeast China (land-2300733) ". We thank the editors and reviewers for their valuable time and insightful comments. These comments are very helpful for us to improve the quality of the manuscript. We carefully studied the comments and revised the manuscript one by one based on constructive comments. The main corrections to the paper and the responses to the reviewers' comments are detailed as follows in this file.
